# The global apparel industry is a significant yet overlooked source of plastic leakage

Anna Kounina [1,8] ✉, Jesse Daystar[2,3,8], Sophie Chalumeau[1], Jon Devine [2], Roland Geyer [4], Steven T. Pires [2,6], Shreya Uday Sonar[5], Richard A. Venditti[6] & Julien Boucher [7]

Plastic pollution is a global environmental threat with potentially irreversible impacts on aquatic life, ecosystems, and human health. This study is a comprehensive assessment of the global apparel industry's contribution to plastic pollution. It includes plastic leakage of packaging and end-of-life apparel waste in addition to fiber emissions during apparel production and use. We estimate that the apparel industry generated 8.3 [4.8–12.3] million tons (Mt) of plastic pollution in 2019, corresponding to 14% [5.5%–30%] of the estimated 60 Mt from all sectors. In this study, we demonstrate that the main source of plastic pollution from the apparel supply chain is synthetic clothing as mismanaged waste either in the country of its original use or in the countries receiving used apparel exports. A fundamental transformation of the apparel economy towards a circular framework and decreased synthetic apparel consumption is needed to tackle apparel-related plastic pollution.

Plastic pollution is a global environmental threat with potentially irreversible impacts on aquatic life, ecosystems, and human health[1–4]. Recent estimates suggest that the planetary boundary for chemical pollution including plastics has already been exceeded. The safe operating space of the planetary boundaries for novel entities is defined to be exceeded when annual production and releases increase at a pace that outstrips the global capacity for assessment and monitoring[5]. As a key piece of the plastic pollution puzzle, primary microplastic emissions into waterbodies receive wide scientific and public attention[6–8]. Microplastics are defined as plastics less than 5 mm. In particular, fiber shedding during the laundering of synthetic textiles has been identified as a key contributor to microplastic release into waterbodies, representing from 4% to 35%[9–12] of all primary microplastic emissions. However, there are missing data pieces in this puzzle. For instance, the amount of microfibers emitted to air as a consequence of wearing clothes[13,14] and drying laundry[15,16] has not yet been quantified. Another missing data point is macroplastic leakage during the disposal of synthetic apparel waste[17]. With rising demand

from a growing middle-income households[18] (i.e. with an annual household income that is two-thirds to double the national median income[19]) across the globe with higher disposable income and the emergence of the 'fast fashion' phenomenon resulting in increased per capita sales in mature economies, clothing production has doubled in the last 15 years[20]. This makes associated macro- and microplastic leakage a potentially growing source of plastic pollution, yet mostly overlooked. Plastic leakage is defined as the quantity of plastic leaving the human-controlled environment and entering the natural environment. This includes littering, intentional or illegal dumping, accidental release, as well as not adequately managed landfills. Plastic leakage is the cause of plastic pollution.

Here, we present a comprehensive assessment of plastic pollution from the global apparel industry through a plastic leakage assessment over the complete apparel life cycle, i.e. from production to manufacturing, use, second life, and disposal. The scope of this study covers global apparel consumption in 2019, including all fiber types. The apparel industry is chosen as a focus, as it is the primary application for

[1]Quantis Switzerland, Rue de la Gare de Triage 5, 1020 Renens, Switzerland. [2]Cotton Incorporated, 6399, Weston Parkway, Cary, NC 27513, USA. [3]Nicholas School of the Environment, Duke University, Durham, NC 27708, USA. [4]Bren School of Environmental Science and Management, University of California, Santa Barbara, CA 93106, USA. [5]Quantis United States, 66 Long Wharf, 2-West, Boston, MA 02110, USA. [6]Department of Forest Biomaterials, College of Natural Resources, North Carolina State University, Raleigh, NC 27695, USA. [7]EA - Earth Action, Chemin des vignes d'argent 7, 1004 Lausanne, Switzerland. [8]These authors contributed equally: Anna Kounina, Jesse Daystar. ✉e-mail: Kounina.anna@quantis.com

textiles. A compilation of textile import data from Trade Data Monitor for the EU, US and Japan revealed that more than 70% of the textiles imported in these markets is apparel. To understand where plastic waste and leakage occur, we study the geospatial flows of new and used apparel based on global apparel consumption in 2019. Our scope includes synthetic, cotton, and all other fibers, which respectively account for 52%, 23%, and 25% of global fiber production[21]. Other fibers include manmade cellulosic fibers (viscose, acetate, etc.), animal fibers (wool, silk, etc.), and plant fibers other than cotton. We separately track seven of the largest markets for new clothes, representing about 70%[22] of the world's apparel consumption (this was derived based on cotton global mill-use data compared against the mill-use consumption for the specific markets assessed. The share of consumption of synthetic apparel across geographies was assumed to be similar to that of cotton apparel): the United States (U.S.), the European Union (EU-28) high and low income countries, Japan, China, India, and Brazil. The European Union (EU-28 in 2019) high and low income countries were defined based on their GDP from World Bank World Development Indicators[23]. The EU28 high income group includes: Luxembourg, Ireland, Denmark, Sweden, Netherlands, Finland, Austria, Belgium, Germany, United Kingdom, France, Italy, Malta and Ciprus. The EU28 low income group includes: Spain, Slovenia, Estonia, Czech Republic, Portugal, Lithuania, Slovakia, Latvia, Greece, Hungary, Poland, Croatia, Romania and Bulgaria. An eighth category, called Rest of World (RoW), covers all remaining markets. In addition, secondary markets importing used apparel from primary markets are accounted for using trade data[24]. We estimate macroplastic losses during waste disposal using the Mismanaged Waste Index (MWI). The MWI is a country-specific ratio of uncollected and improperly managed (not recycled, incinerated, or disposed of in a properly managed landfill) waste over total waste generation, calculated based on Kaza et al.[25] for most countries

and refined data for the U.S., the E.U. high and low income, China, India, Pakistan, and Brazil. We provide plastic waste and leakage results as midpoint values with a confidence interval to reflect the uncertainty of apparel consumption data and the MWI. More details on the plastic leakage methodology, trade data, textile-specific MWI calculation and the uncertainty analysis can be found in Methods.

## Results
### Global plastic waste generation and leakage
Global apparel consumption in 2019 is estimated as 32 [31 – 33] million tons (Mt), split between synthetic apparel (15 [14–15] Mt), cotton apparel (15 Mt), and other fibers (2.6 Mt) (Supplementary Table 1). The difference in fiber type proportions between global production and apparel-specific consumption is due to the fact that cotton fibers are used mainly for apparel. In contrast, synthetic fibers are used for a wide range of applications, from automobile upholstery to stuffed toys. It is estimated that the global apparel sector generated 21 [20–22] Mt of plastic waste in 2019. Virtually all of it was macroplastic waste. The apparel value chains of synthetic, cotton, and other fibers generated macroplastic waste of 18, 1.9, and 0.31 Mt/year, respectively (Fig. 1). The synthetic apparel value chain is thus responsible for 89% of macroplastic waste from the apparel industry, with the cotton and other fibers value chains contributing 9.4% and 1.5%, respectively. Plastic waste from synthetic apparel is dominated by the end-of-life textile itself, with packaging contributing only a small portion to the total. Synthetic end-of-life apparel makes up 81% of the total plastic waste from the global apparel industry. Plastic waste from the cotton apparel value chain is driven by packaging, with a small additional contribution from plastic mulching on cotton fields in some geographies. The plastic waste created by the other fibers value chains is almost entirely due to packaging.

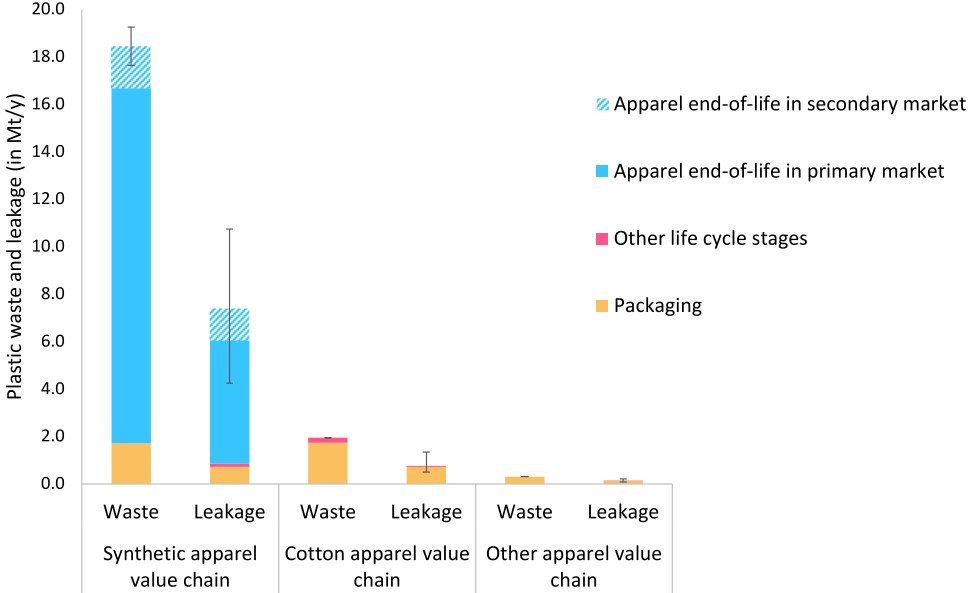

**Fig. 1 | Overview of apparel macroplastic waste and macro- and microplastic leakage.** The plastic waste along the apparel value chain includes macroplastics disposed of during agricultural activities; yarn, fabric, and apparel production sites; retail and consumer use; end-of-life, second life, second end-of-life, as well as transport. The plastic leakage into the environment represents the mass of macroplastics (from product and packaging mismanaged waste) and microplastics (from plastic pellets and microfiber release, as well as tire abrasion during transport) released in freshwater, oceans, terrestrial environment, and soils from the apparel value chain. We provide plastic waste and leakage results as midpoint values with a confidence interval to reflect the uncertainty of the consumption numbers for China and Rest of the World presented in Supplementary Table 2, and

the uncertainty of the Mismanaged Waste Index (MWI) for the synthetic apparel value chain. The approach chosen to calculate the uncertainty for textile specific MWIs is based on the evaluation of the standard deviation in MWI values for general waste obtained through different methodologies for the same country and is presented in Supplementary Table 15. For packaging and export countries MWI, we took the uncertainty range of the packaging MWI as the interval between the 1st and the 3rd quartile of the MWI of the economic group they belong to: High income countries (HIC), Upper middle income countries (UMC), Lower middle income countries (LMC), Low income countries (LIC), presented in Supplementary Table 17. Source data are provided as a Source Data file.

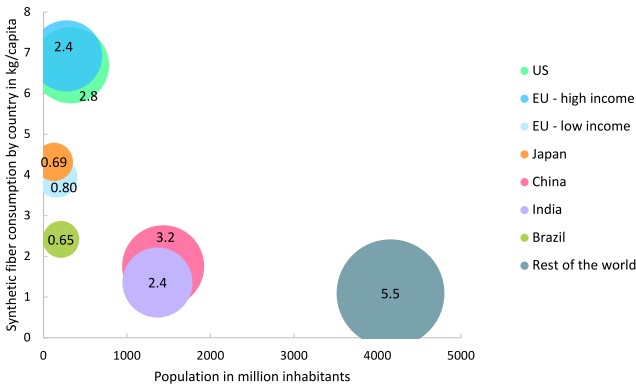

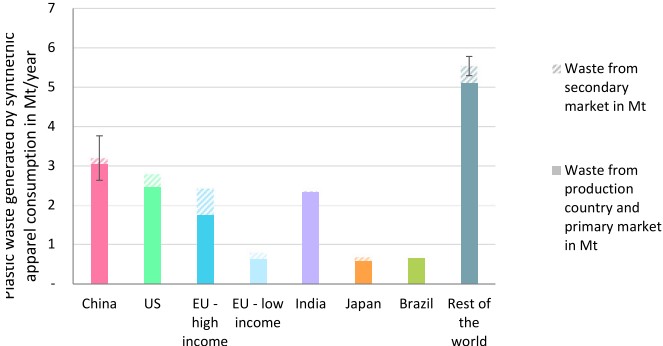

A

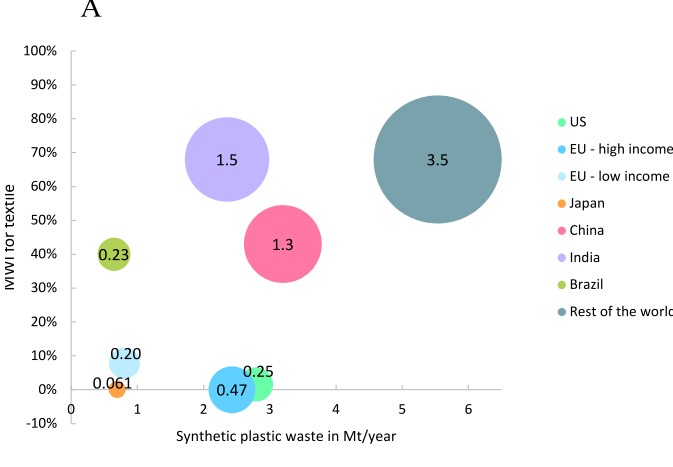

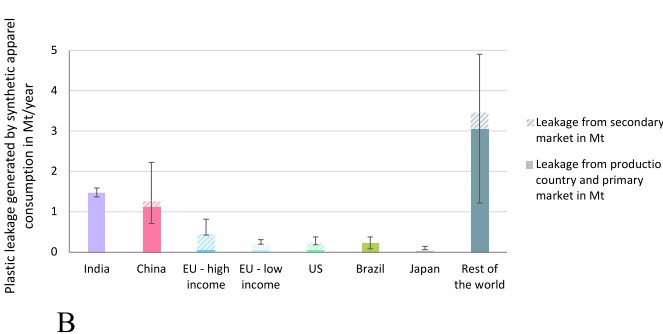

B

**Fig. 2 | Plastic waste and plastic leakage in Mt/year generated by synthetic apparel consumption in EU28 high and low income countries, U.S., Japan, China, India, Brazil, and the Rest of the World in primary and secondary markets.** Plastic waste is expressed as a function of population and apparel consumption per capital (**A**) and plastic leakage – as a function of plastic waste and country specific Mismanaged Waste Index (MWI) (**B**). In (**A**, **B**), the size of the pie chart represents respectively the amount of plastic waste and leakage. As in Fig. 1, we provide plastic waste and leakage results as midpoint values with a confidence interval to reflect the uncertainty of the consumption numbers for China and Rest of the World presented in Supplementary Table 2, and the uncertainty of the MWI for the synthetic apparel value chain. Source data are provided as a Source Data file.

primary and secondary markets is by far the largest contributor with 6.6 [4.1–10] Mt/year. It makes up 88% of the total plastic leakage from the synthetic apparel value chain and 79% of the leakage from the entire global apparel sector. The release of synthetic microfibers during apparel washing is included in the "Other life cycle stages" category and represents less than 1.5% of total plastic leakage from the apparel industry. This study does not consider non-synthetic microfiber emissions as they are not plastic. Plastic leakage from mismanaged packaging is estimated to be 0.71, 0.71, and 0.12 Mt/year from the apparel life cycles of synthetic, cotton, and other fibers, respectively. The plastic leakage estimate of the cotton value chain also has a small contribution from plastic mulch losses during cotton cultivation. All calculations from raw data to final results are available in Supplementary Data 1.

### Synthetic apparel plastic leakage by geography (including packaging)

We further analyze the value chain for synthetic apparel, which is the main driver of plastic leakage from apparel (7.4 Mt/year), by geographic market. For the end-of-life, the primary market, where new apparel is originally sold and used, is distinguished relative to the secondary market, to which some apparel is exported after the end of its first use. Throughout this study, the waste generation and subsequent leakage occurring in secondary markets are attributed to the primary market where the apparel was originally sold. Double counting is avoided as these results focus on waste and leakage generation after sorting for export in a second-hand market.

Figure 2A shows plastic waste generation of the synthetic apparel industry, which is driven by a combination of population size and consumption per capita. In 2019, the primary apparel markets of Rest of the World, China, the U.S., EU28 – high income, and India generated 5.5 [5.3–5.8] Mt, 3.2 [2.6–3.8] Mt, 2.8 Mt, 2.4 Mt, and 2.4 Mt of plastic waste respectively. Population size is a key driver for China, India, and the Rest of the World. For the U.S., Europe, and Japan – the driver is apparel consumption per capita. The share of waste sent to export markets always represents less than a third of the total plastic waste generated in each market.

We then show plastic leakage for the same geographies (Fig. 2b Supplementary Table 17) as a function of the plastic waste weight and MWI in the primary market. India, China, and the Rest of the World are key contributors to plastic leakage, accounting for 1.5 [1.4–1.6] Mt/year, 1.3 [0.71–2.2] Mt/year, and 3.5 [1.2–4.9] Mt/year of synthetic apparel leakage, respectively. This is due to high textile MWI in the primary markets (68% of plastic waste is mismanaged in India, 68% in RoW, and 43% in China).

For apparel originally sold in the EU28 high income countries, Japan, the U.S. and EU28 low income countries, 93%, 80%, 80%, and 77% of the plastic leakage due to mismanaged synthetic apparel waste occurs in second-hand export countries, respectively. This means that apparel consumption in these geographies generates more plastic pollution in the secondary markets that receive their used apparel exports than in their own primary markets.

The fraction of plastic waste generated by the global apparel sector in 2019 that ultimately leaked into the environment from all fibers is estimated to be 8.3 [4.8–12.3] Mt/year, corresponding to 40% [22%–62%] of the plastic waste generation. 89% of this stems from the life cycle of synthetic apparel, while 9.2% and 1.5% are from the apparel life cycles of cotton and other fibers. Synthetic apparel end-of-life in

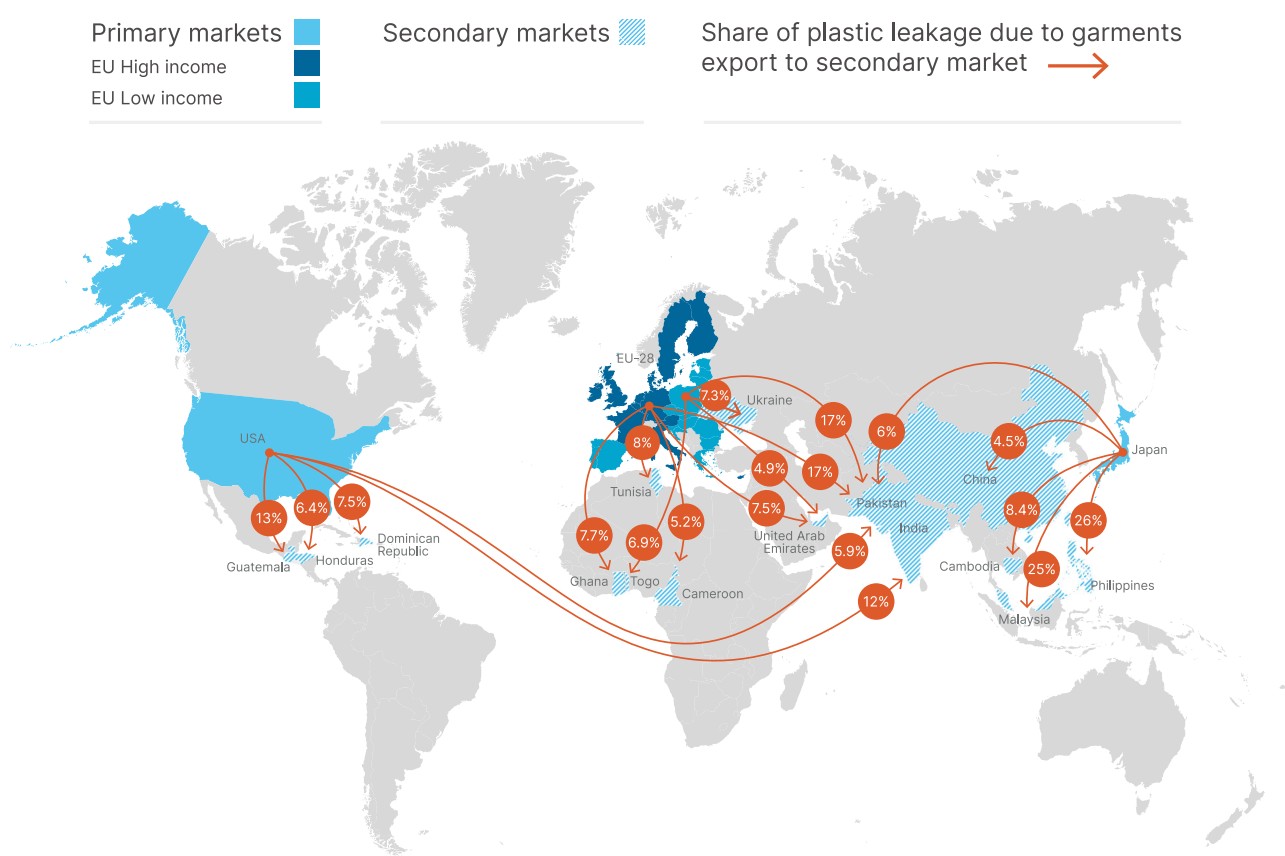

**Fig. 3 | Influence of second-hand apparel exports on the location of synthetic apparel plastic leakage from EU28 high and low income countries, U.S., and Japan.** Trade data[21] has been used to quantify the leakage in secondary markets from imported second-hand textiles. Arrows show all export countries for which the plastic leakage is larger than 5% of the primary market's total leakage. Source data are provided as a Source Data file.

## Synthetic apparel plastic leakage in secondary markets

The total leakage of synthetic apparel end-of-life in secondary markets is estimated to be 1.4 [1.2 – 1.6] Mt/year, or 17% of the total plastic leakage from the global apparel industry. Figure 3 and Table 1 illustrate the relationship between primary markets and plastic leakage in their secondary markets. It shows key secondary markets, which are defined as having an associated plastic leakage larger than 5% of the primary market's total leakage. Most export countries listed in Table 1 vary depending on the market geographies, although Pakistan is a key export market for all geographies. These secondary markets are based on trade data[24] and represent the first country of export, while the end destination market can be different.

## Discussion

The plastic leakage generated in 2019 by the global apparel industry ranges from 4.8 to 12.3 Mt, with a midpoint estimate of 8.3. This is 14% [5.5% –30%] of total annual plastic leakage, which we estimate as 60 [41–87] Mt/year. This total plastic leakage consists of 41 Mt/year [29–60] of macroplastic from packaging (including dumpsites, aquatic and terrestrial pollution)[26], 3.2 Mt/year [1.8–5.0] of primary microplastic[10], 7.4 Mt/year [5.2–9.8] macro and micro-plastics from paint[11] and the 8.3 [4.8 – 12.3] Mt/year of macroplastic waste generated by the apparel industry. The latter is dominated (on a mass basis) by mismanaged synthetic apparel waste after its first or second life. By mass (unlike by the number of particles released), primary microfiber releases during synthetic fiber production and apparel washing (during both manufacturing and use stages) represent less than 1.5%, i.e. 0.11 Mt/year of plastic leakage from the global apparel industry.

As used synthetic apparel is exported to secondary markets, plastic waste generation is shifted from geographies with robust waste management systems to geographies with inadequate waste collection and treatment. It is thus vital that textile brands and governments include secondary markets in their sustainability strategies, roadmaps, and action plans to limit their plastic pollution footprint.

Despite recent improvements in data availability, assessments similar to the one herein are still subject to significant methodology and data gaps. First, current research is not mature enough to adequately convert masses and types of plastic leakage into impacts on ecosystems and humans. The number of plastic particles/fibers may be as relevant an indicator as mass[27,28]. Impact frameworks[29] and methods[30,31] are currently being developed to address this methodological gap. When available, such methodologies should be integrated with plastic leakage assessments to better understand the impacts related to each plastic type and size. Second, the most important parameter affecting macroplastic leakage results is the country-specific MWI. While this study developed seven textile-specific MWIs for China, India, Pakistan, EU28 high income and EU28 low income countries, Brazil, and the U.S. to reduce uncertainty, global databases such as Plasteax[32] need to be complemented with a wider coverage to provide a more reliable basis for future plastic leakage assessments. PLASTEAX is a global data platform dedicated to plastic environmental analytics, providing access to polymer and application specific waste management & leakage data. Third, there is no peer-reviewed publication that contains estimates for end-use apparel consumption by fiber type and region. Figures for apparel consumption in the U.S., the EU-28 high income and low income countries, China, India, Japan, Brazil, and the aggregated Rest of the World have been compiled

**Table 1 | Summary data of second-hand apparel exports for the location of synthetic apparel plastic leakage of EU28, U.S., and Japan**

| Primary market | Total leakage associated with the primary market (Fig. 2B) | Sum of leakage in all secondary markets | Key secondary markets | MWI of the secondary market, % | Leakage generated in the secondary market | |
|---|---|---|---|---|---|---|
| | in Mt | in Mt | | | in Mt | in % of total leakage associated with the primary market |
| E.U. High income | 0.47 | 0.44 | Pakistan | 98 | 0.077 | 17 |
| | | | Tunisia | 73 | 0.037 | 8 |
| | | | Ghana | 89 | 0.036 | 7.7 |
| | | | United Arab Emirates | 70 | 0.035 | 7.5 |
| | | | Cameroon | 88 | 0.024 | 5.2 |
| | | | Other markets | - | 0.23 | 53 |
| E.U. Low income | 0.20 | 0.16 | Pakistan | 98 | 0.034 | 17 |
| | | | Ukraine | 60 | 0.015 | 7.3 |
| | | | Togo | 99 | 0.014 | 6.9 |
| | | | United Arab Emirates | 70 | 0.0099 | 4.9 |
| | | | Other markets | - | 0.088 | 55 |
| US | 0.25 | 0.21 | Guatemala | 82 | 0.031 | 13 |
| | | | India | 68 | 0.029 | 12 |
| | | | Dominican Republic | 94 | 0.018 | 7.5 |
| | | | Honduras | 73 | 0.016 | 6.4 |
| | | | Pakistan | 98 | 0.015 | 5.9 |
| | | | Other markets | - | 0.098 | 47 |
| Japan | 0.061 | 0.052 | Philippines | 85 | 0.016 | 26 |
| | | | Malaysia | 22 | 0.015 | 25 |
| | | | Cambodia | 100 | 0.0052 | 8.4 |
| | | | Pakistan | 98 | 0.0037 | 6 |
| | | | China | 43 | 0.0028 | 4.5 |
| | | | Other markets | - | 0.0089 | 17 |

based on production, import, and export estimates or extrapolations from other fibers. Fourth, trade data covering used textile exports[24] does not necessarily reflect the final destination. Last, this study includes releases of microfibers to water from textile washing, but microfiber emissions to air are excluded due to a lack of data[13,14]. Further experimental studies should be performed to better understand the magnitude of microfiber emissions into the air as part of the broader problem of plastic leakage in the apparel value chain. Finally, the only plastic flow we were able to account for in the category "other fibers" was apparel packaging.

Current efforts to transform the global plastics economy and meaningfully reduce plastic emissions[33] formalized by the development of legally binding U.N. Treaty on plastic pollution[34], need to include the apparel sector in addition to packaging and particularly focus on the end-of-life disposal of synthetic apparel. The apparel industry should increase its efforts to work towards true circularity and reduce overconsumption, following ambitions such as phasing out substances of concern and synthetic microfiber release, increasing clothing lifetime utilization, radically improving apparel waste management and recycling, making effective use of resources and move to renewable inputs[20]. Key levers to achieve such a systemic transition include the design of clothing for durability, reuse, remanufacturing, and recycling to keep products and materials in the economy. Additional needs include improving waste management infrastructure for textiles, i.e. developing and scaling processing and recycling technologies, especially in developing countries. The use of natural fibers is another readily available option to reduce plastic leakage from the apparel industry. With any of these actions, sustainability metrics, in addition to plastic leakage, need to be considered in order to avoid unintended consequences and burden shifting. Our results demonstrate that synthetic end-of-life apparel generates 14% [5.5%–30%] of the total plastic pollution due to insufficient waste management infrastructure in their primary and secondary markets. Without a systemic transition in apparel production, consumption, and disposal, the forecasted increases in synthetic fiber production and use[35] will contribute substantially to the ongoing growth in plastic pollution.

## Methods

### Estimation of consumption per market

We estimate the total mass of apparel consumed in each market studied based on a combination of production and import data.

For cotton apparel, we calculate net apparent consumption data by combining data on domestic production plus imports minus exports. We extract the quantity of cotton for mill-use (e.g., yarn production) for each market from the USDA[36], then we add imports and subtract exports at each downstream stage of the value chain (fabric production and end-product manufacturing) based on trade data[24]. We account for losses throughout the value chain using the conversion factors established by the USDA.

For synthetic apparel, the approach based on net apparent consumption cannot be used due to the wider range of end-uses relative to cotton, which distort the derivation of apparel estimates. Therefore, we derive the synthetic consumption numbers for the U.S., EU-28 high and low income countries, and Japan based on apparel import data and

**Table 2 | MWI refined for textiles**

| Market | MWI for textiles (%) | Reference year |
|---|---|---|
| E.U. High income | 0 | 2020 |
| E.U. Low income | 7.8 | 2021 |
| US | 1.6 | 2020 |
| China | 43 | 2019 |
| India | 68 | 2022 |
| Brazil | 40 | 2017 |
| Pakistan | 98 | 2019 |
| Japan | 1.8 | 2019 |

for India, China, and Brazil based on assumptions about cotton's share of apparel consumption. For China, there is uncertainty about the cotton trade numbers due to a lack of harmonization in the reporting. Moreover, synthetic textile production in China is very large, and the diversity of uses for domestic consumption is not reported. For these reasons, we choose to establish an uncertainty range on the apparel consumption for the Chinese market specifically.

Other fibers include both manmade and natural fibers. The manmade other fibers are accounted for in the same way as the synthetic fibers. For the natural fibers, there is a lack of data on production, so consumption shares from U.S. trade data are used and assumed to be representative of natural fiber share of apparel consumption for all markets.

The main countries of export of used apparel at the end-of-life are also compiled for each primary market from trade data[24] (SI: Supplementary Tables 6–12).

More details on the derivations can be found in the supplementary materials.

### Plastic flows along the apparel value chain

We quantify the plastic flows involved at each stage of the cotton, synthetic and other fibers value chain based on previous studies[37–42], from agricultural production or pellet production to manufacturing, use, second life, and eventual disposal (Supplementary Figs. 1–3). We also include the primary and secondary packaging of the apparel. We include the release of microplastics into water from pre-consumer textile manufacturing[43] as well as laundering during use. Microfiber release into air are still poorly covered in the current literature[13,14] and cannot be included because a wider set of studies would be required to derive mean release rates. The fully considered systems and the plastic flows at each stage are detailed in the supplementary materials.

At the end-of-life, some apparel is recycled, landfilled, or incinerated within the same country, whereas some are exported to other countries to be reused and eventually disposed of. To cover apparel exports, we extracted the share of textiles exported at the end-of-life (trade code HS 6309) over total textile consumption in the market from trade data (Supplementary Tables 6–12).

### Estimation of the leakage of macro- and micro-plastics at each step

At each step of the apparel value chain, we assess the macro- and micro-plastics leakage by combining the plastic flows with the associated loss and release rates into the environment[42].

These include plastic pellet losses during manufacturing, microplastic fiber losses through laundering, microplastic release through wastewater, macroplastic losses during waste disposal, and microplastic losses and release through transport. Macroplastic losses during waste disposal are driven by the Mismanagement Waste Index (MWI), a country-specific rate of the plastic waste not adequately treated (not recycled, incinerated or disposed of in a properly managed landfill), calculated according to Peano et al.[42]. For the countries identified as apparel leakage hotspots (the U.S., the E.U. high and low income, China,

India, Pakistan, and Brazil), the MWI is refined to represent textile waste specifically with 2018-2019 data (Table 2, Supplementary Fig. 5).

Since the MWI greatly influences the overall quantity of plastic leaked and has a high uncertainty associated, we perform an uncertainty analysis on these numbers (Supplementary Tables 15, 16 and 17).

### Data availability

The calculated data used in this study (e.g. apparel manufacturing by country, used apparel export) are provided in the Supplementary Information. Source data are provided with this paper. For ref. 21 Trade Data Monitor. Trade Data Monitor. (2020). Available at: https://tradedatamonitor.com/. The search terms included all six-digit trade classification codes (commodities) available under Harmonized System (HS) Chapters 50–63. The four-digit HS code 6309 was used to access figures for used textiles. The trade partner parameter was set to the "World" or "All countries". Data were downloaded by country of interest for 2019. The dates of access were in 2021. For SI ref. 1 United States Department of Agriculture, F. A. S. Production Supply and Distribution. Available at: https://apps.fas.usda.gov/psdonline/app/index.html#/app/advQuery. The search terms are Cotton, Domestic Use, and World Total. Crop data from the USDA are published by crop years that run from August through July. The crop year that was used was 2018/19. The dates of access were in 2021. Should any raw data files be needed in another format they are available from the corresponding author upon reasonable request. Source data are provided with this paper.

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

## Acknowledgements

We thank D. Boyer, A. Ernstoff, S. Humbert, M. Haydn, M. Palacios and M. Gallato for input and comments on earlier drafts as well as figure design.

## Author contributions

Conceptualization: A.K., J.Da., S.C., S.T.P., S.U.S., J.B.; Data curation: J.De., S.C.; Formal analysis: S.C., A.K.; Funding acquisition: J.Da, S.T.P.: Investigation: A.K., J.Da., S.C., J.De., R.G., S.T.P., S.U.S., R.A.V., J.B.; Methodology: A.K., J.Da., S.C., J.De., R.G., S.T.P., S.U.S., R.A.V., J.B.; Project administration: A.K., S.C.; Supervision: J.B., R.G.; Writing – original draft: A.K., J.Da., S.C., J.De., R.G., S.T.P., S.U.S., R.A.V., J.B.; Writing – review & editing: A.K., J.Da., S.C., J.De., R.G., S.T.P., S.U.S., R.A.V., J.B.

## Competing interests

The authors declare no competing interests.

## Ethical approval

The authors followed the recommendations set out in the Global Code of Conduct for Research in Resource-Poor Settings (https://www.globalcodeofconduct.org/) whenever applicable.
