## [Peer Review File · Nature Communications]

REVIEWER COMMENTS

Reviewer #1 (Remarks to the Author):

The study is interesting and well done. The study makes a clear contribution to this knowledge field and provides totally new information from plastics leakage from the textile system.

Paper meets all requirements for academic study and publication.

The paper does not need any rewriting or corrections.

Reviewer #2 (Remarks to the Author):

Dear authors,

Many thanks for an interesting paper. I have enjoyed the perspective given to the paper and I understand the huge limitations existing when trying to bring together a paper of these characteristics in terms of data quality. However, a recent study by Alencar and colleagues (2022) - (<https://doi.org/10.1016/j.jenvman.2022.116195>) provides some interesting data on how to improve mismanaged waste data from countries in the Global South. I would recommend that your study integrates this information as a way to expand the level of resolution of your estimates. This would be my main suggestion to improve the data management you have performed.

Reviewer #3 (Remarks to the Author):

The authors aim to provide an assessment of the global apparel industry's contribution to plastic pollution. The authors reveal the related knowledge gaps in the current literature, however, this global assessment relies on many secondary data from various sources and assumptions to fill the lack of data in equations provided in SI materials.

These are recognized by authors, particularly in the supplemental info file. Detailed explanations and literature sources are mentioned here.

The Waste Mismanagement Index is the key indicator to determine the plastic leakage source to the natural environment derived from MSW disposal routes (e.g What a Waste 2.0 and applied to textile/apparel waste flows). This indicator is problematic to assess at the global level and to compute for each country – because the national level data are inconsistent, outdated for some countries, and ignore the subnational variations (e.g urban vs rural regions).

Despite the limitation of data, this paper draws attention to the plastic leakage of all life-cycle associated with the apparel industry that could stimulate further investigations to improve the current models and input data

The authors selected the 6 most important markets + rest of the world as geospatial analyses for apparel industries. However, EU-28 is examined as one geographical area (market) besides Japan for example. This is a bit problematic because there are 28 countries with significant disparities in waste management performances confirmed by previous studies (& Eurostat data) that will affect the plastic leakage flow. This should be stipulated as a research limitation in the manuscript text.

Both macroplastic and microplastic leakage is considered in the manuscript. The role of import-export flows is examined and this paper shows the role of secondary markets in the plastic leakage problem.

Introduction – should be expanded and have a broader view of textile waste management problems --to point out the role of fast fashion (keyword missing in the manuscript) in overconsumption and textile waste production.

More discussion around the figures should be further developed.

The main manuscript text should include more aspects regarding the research limitations (described in the supplemental materials).

No mention of the circular fashion in the discussion section – this part of the manuscript needs a broader analysis. The word "circular framework" is mentioned in the abstract but not in the main body text –, particularly in the " A need for a systemic transformation " section

Overall, I suggest that some info provided in SI materials (research limitations, methods, circular economy in the apparel industry) should be included in the manuscript text for better clarity and context/background support.

It will difficult for the audience to read and follow the main text flow with so many redirections to SI materials.

Dear reviewers,

Thank you for your diligent work. We provided an answer to each of your comments (see below) and highlighted the corresponding modifications in the article text **in yellow**. Shall you have any remaining issues, please do not hesitate to come back to us.

With our kind regards

REVIEWER COMMENTS

Reviewer #1 (Remarks to the Author):

The study is interesting and well done. The study makes a clear contribution to this knowledge field and provides totally new information from plastics leakage from the textile system. Paper meets all requirements for academic study and publication. The paper does not need any rewriting or corrections.

Answer 1.1: Thank you for this comment.

Reviewer #2 (Remarks to the Author):

Dear authors,

Many thanks for an interesting paper. I have enjoyed the perspective given to the paper and I understand the huge limitations existing when trying to bring together a paper of these characteristics in terms of data quality. However, a recent study by Alencar and colleagues (2022) - (<https://doi.org/10.1016/j.jenvman.2022.116195>) provides some interesting data on how to improve mismanaged waste data from countries in the Global South. I would recommend that your study integrates this information as a way to expand the level of resolution of your estimates. This would be my main suggestion to improve the data management you have performed.

Answer 2.1: Thank you for referring to the work performed by Alencar et al. Although their approach is very relevant and granular, we could not extend it to the geographies covered in our manuscript beyond Brazil. We updated the MWI for all geographies using a refined methodology and sources presented in Supplementary Fig. 6 and Table 14.

Reviewer #3 (Remarks to the Author):

The authors aim to provide an assessment of the global apparel industry's contribution to plastic pollution. The authors reveal the related knowledge gaps in the current literature, however, this global assessment relies on many secondary data from various sources and assumptions to fill the lack of data in equations provided in SI materials. These are recognized by authors, particularly in the supplemental info file. Detailed explanations and literature sources are mentioned here.

The Waste Mismanagement Index is the key indicator to determine the plastic leakage source to the natural environment derived from MSW disposal routes (e.g. What a Waste 2.0 and applied to textile/apparel waste flows). This indicator is problematic to assess at the global level and to compute for each country – because the national level data are inconsistent, outdated for some countries, and ignore the subnational variations (e.g. urban vs rural regions).

Despite the limitation of data, this paper draws attention to the plastic leakage of all life-cycle associated with the apparel industry that could stimulate further investigations to improve the current models and input data

The authors selected the 6 most important markets + rest of the world as geospatial analyses for apparel industries. However, EU-28 is examined as one geographical area (market) besides Japan for example. This is a bit problematic because there are 28 countries with significant disparities in waste management performances confirmed by previous studies (& Eurostat data) that will affect the plastic leakage flow. This should be stipulated as a research limitation in the manuscript text.

Answer 3.1: We update MWIs as shown in Supplementary Table 14 and summarized in Table 3.1 below, based on more comprehensive sources specific to textiles for EU High GDP, EU Low GDP, US, China, India, Brazil, Pakistan and Japan. We disaggregated Europe in the High and Low GDP group to reduce the MWI variability in each of these markets.

The MWI was overestimated for Europe (29%) and decreased as we updated with data from *Watson et al. (June 2020). Towards 2025: Separate collection and treatment of textiles in six EU countries* to 0% for high income countries and 7.8% for low income countries.

The MWI for India also decreased from 98% to 68%.

Table 3.1: Updated MWI

Market	MWI for textiles (%)	Reference year	Updated MWI for textiles (%)	Updated reference year
EU-28	29%	2018	EU High Income: 0% EU Low Income: 7.8%	2020 2021
US	2.7%	2018	1.6%	2020
China	45%	2018	43%	2019
India	98%	2018	68%	2022
Brazil	45%	2019	40%	2017
Pakistan	100%	2016	98%	2019
Japan			1.8%	2019

Both macroplastic and microplastic leakage is considered in the manuscript. The role of import-export flows is examined and this paper shows the role of secondary markets in the plastic leakage problem.

Introduction – should be expanded and have a broader view of textile waste management problems -- to point out the role of fast fashion (keyword missing in the manuscript) in overconsumption and textile waste production.

Answer 3.2: We expanded the introduction with the following sentence at line 31:

“With rising demand from a growing middle class across the globe with higher disposable income and the emergence of the 'fast fashion' phenomenon, clothing production has doubled in the last 15 years

¹⁸, making associated macro- and microplastic leakage a potentially growing source of plastic pollution, yet mostly overlooked.”

More discussion around the figures should be further developed.

Answer 3.3: We included additional interpretation for Figure 1, 2 and 3, respectively:

At line 71:

“It is estimated that the global apparel sector generated 21 Mt of plastic waste in 2019. Virtually all of it was macroplastic waste.”

At line 78:

“Synthetic end-of-life apparel makes up 81% of the total plastic waste from the global apparel industry.”

At line 82:

“The fraction of plastic waste generated by the global apparel sector in 2019 that ultimately leaked into the environment from all fibers is estimated to be 8.4 [4.8 – 12.2] Mt/year, corresponding to 40% [22% - 62%] of the plastic waste generation. 89% of this stems from the life cycle of synthetic apparel, while 9.4% and 1.5% are from the apparel life cycles of cotton and other fibers. Synthetic apparel end-of-life in primary and secondary markets is by far the largest contributor with 6.6 [4.1– 10] Mt/year. It makes up 89% of the total plastic leakage from the synthetic apparel value chain and 79% of the leakage from the entire global apparel sector. The release of synthetic microfibers during apparel washing is included in the “Other life cycle stages” category and represents less than 1.5% of total plastic leakage from the apparel industry. This study does not consider non-synthetic microfiber emissions as they are not plastic. Plastic leakage from mismanaged packaging is estimated to be 0.71, 0.71, and 0.12 Mt/year from the apparel life cycles of synthetic, cotton, and other fibers, respectively. The plastic leakage estimate of the cotton value chain also has a small contribution from plastic mulch losses during cotton cultivation.”

At line 121:

“The share of waste sent to export markets always represents less than a third of the total plastic waste generated in each market.”

At line 166:

“Most export countries listed in Table 1 vary depending on the market geographies, although Pakistan is a key export market for all geographies. These secondary markets are based on trade data ²¹ and represent the first country of export, while the end destination market can be different.”

The main manuscript text should include more aspects regarding the research limitations (described in the supplemental materials).

Answer 3.4: We updated the limitation paragraphs in the discussion as follows at line 196:

“Despite recent improvements in data availability, assessments similar to the one herein are still subject to significant methodology and data gaps. First, current research is not mature enough to adequately convert masses and types of plastic leakage into impacts on ecosystems and humans. The number of plastic particles/fibers may be as relevant an indicator as mass ^{24,25}. Impact frameworks ²⁶ and methods ^{27,28} are currently being developed to address this methodological gap. When available, such methodologies should be integrated with plastic leakage assessments to better understand the impacts

related to each plastic type and size. Second, the most important parameter affecting macroplastic leakage results is the country-specific MWI. While this study developed seven textile-specific MWIs for China, India, Pakistan, EU28 high income and EU28 low income countries, Brazil, and the U.S. to reduce uncertainty, global databases such as Plasteax^{1 29} need to be complemented with a wider coverage to provide a more reliable basis for future plastic leakage assessments. Third, there is no peer-reviewed publication that contains estimates for end-use apparel consumption by fiber type and region. Figures for apparel consumption in the U.S., the EU-28 high income and low income countries, China, India, Japan, Brazil, and the aggregated Rest of the World have been compiled based on production, import, and export estimates or extrapolations from other fibers. Fourth, trade data covering used textile exports²¹ does not necessarily reflect the final destination. Last, this study includes releases of microfibers to water from textile washing, but microfiber emissions to air are excluded due to a lack of data^{13,14}. Further experimental studies should be performed to better understand the magnitude of microfiber emissions into the air as part of the broader problem of plastic leakage in the apparel value chain. Finally, the only plastic flow we were able to account for in the category "other fibers" was apparel packaging."

No mention of the circular fashion in the discussion section – this part of the manuscript needs a broader analysis. The word "circular framework" is mentioned in the abstract but not in the main body text –, particularly in the "A need for a systemic transformation" section

Answer 3.5: We updated the following sentence at line 223:

"The apparel industry should increase its efforts to work towards true circularity and reduce overconsumption, following ambitions such as phasing out substances of concern and synthetic microfiber release, increasing clothing lifetime utilization, radically improving apparel waste management and recycling, making effective use of resources and move to renewable inputs¹⁸."

Overall, I suggest that some info provided in SI materials (research limitations, methods, circular economy in the apparel industry) should be included in the manuscript text for better clarity and context/background support.

It will difficult for the audience to read and follow the main text flow with so many redirections to SI materials.

Answer 3.5: We included the key explanations from the SI in the Method section, e.g. Table 2. As explained in answer 3.4, we expanded the discussion on limitations.

Regarding the circular economy in the apparel industry, this has been widely described in one cited reference *EMF (2020) A new textiles economy: Redesigning fashion's future*. The goal of this manuscript is to provide a quantification of the plastic leakage from the apparel industry and to provide outlooks on potential solutions, while not entering in an in-depth discussion on the implementation of a circular economy in the apparel industry,

¹ PLASTEAX is a global data platform dedicated to plastic environmental analytics, providing access to polymer and application specific waste management & leakage data.

REVIEWERS' COMMENTS

Reviewer #1 (Remarks to the Author):

The corrections have improved the paper.

Reviewer #3 (Remarks to the Author):

The current version of the manuscript is much improved than the initial submission.

The authors respond to my previous queries related to the paper.

The method and results examine the limitations of the study and detailed explanations are provided in the SI materials. Despite those limitations, the paper tries to fill the existing gaps related to plastic leakage flow associated with the global apparel industry.

The authors revised their modeling for some geographies (eg data breakdown to EU high income – EU low income) and adjusted some MWI levels for textiles. The results are provided in midpoints and intervals to reveal the uncertainties related to their assessment.

In my opinion, the paper is acceptable for publication.